# Presence of SARS-CoV-2 RNA in Semen—Cohort Study in the United States COVID-19 Positive Patients

**Bruno Machado [1,*]** , **Gustavo Barcelos Barra [2]** , **Nickolas Scherzer [1]** , **Jack Massey [1]** , **Hemerson dos Santos Luz [3]** , **Rafael Henrique Jacomo [2]** , **Ticiane Herinques Santa Rita [2]** and **Rodney Davis [1]**

1. Urology Department, School of Medicine, University of Arkansas for Medical Sciences, Little Rock, AR 72205, USA; ndscherzer@uams.edu (N.S.); jlmassey@uams.edu (J.M.); rdavis@uams.edu (R.D.)
2. Clinical Analisys Laboratory, Sabin Medicina Diagnostica, Brasilia, DF 70632-340, Brazil; gustavo@sabin.com.br (G.B.B.); rafaeljacomo@sabin.com.br (R.H.J.); ticianehenriques@sabin.com.br (T.H.S.R.)
3. Infectious Disease Division, Hospital das Forças Armadas, Brasilia, DF 70675-731, Brazil; hemersonluz@hotmail.com
* Correspondence: bmachado@uams.edu

**Abstract:** On 31 December 2019, China informed the World Health Organization they were facing a viral pneumonia epidemic of a new type of Coronavirus. Currently, 10 months later, more than 43,000,000 people have been infected, and about 1,150,000 deceased worldwide from the disease. Knowledge about the virus is updated daily, and its RNA was isolated from several human secretions, e.g., throat, saliva, pulmonary alveolar washing, and feces. So far, only one publication found the presence of SARS-CoV-2 in semen. In this 5-month cross-sectional study, we recruited 15 patients diagnosed with a positive nasal swab for SARS-CoV-2 with no or mild symptoms in our institution. A semen sample after a shower was retrieved and tested for viral RNA in the semen. The samples were tested for the viral RNA with RT-PCR with two different genetic probes. The samples were re-tested 24 h after the first test to confirm the results. The SARS-Cov-2 viral RNA was present in 1/15 patients [6.66%] in our sample. Even in a small sample, the RNA from SARS-CoV-2 can be isolated from human semen. This information should alert the scientific community and public health officials about a possible new form of transmission of the disease and long-term clinical effects on the population.

**Keywords:** SARS-CoV-2; Semen; COVID-19

## 1. Introduction

On 31 December 2019, China informed the World Health Organization that they were facing an unknown Coronavirus pneumonia epidemic. The coronavirus is a well-known family of viruses; two of their strains [α and β] can infect humans. The new epidemic's causative agent was named SARS-CoV-2 and is a positive-sense single-strained RNA virus encapsulated by a protein envelope that binds to the ACE2 receptor in the pulmonary cells to enter and replicate [1].

Since global disease spread started in January 2020, over 42,000,000 people have been infected, with more than 1,152,000 casualties [2]. The public health challenges are enormous, and governments all over the world have taken measures to try diminish the virus spreading through their countries [3].

In 80% of the patients, the virus will not cause any symptoms; however, these patients are the ones who represent the most danger since they can silently spread the disease. From the 20% that will present symptoms, 75% will have mild symptoms, and only 25% will need hospital admission for treatment [4].

The most common symptoms from COVID-19 are fever or chills, cough, shortness of breath, fatigue, muscle or body aches, headache, new loss of taste or smell, sore throat, nasal congestion, nausea or vomiting, and diarrhea [5].

The reverse transcriptase PCR assays [RT-PCR] have been used by several authors to isolate the virus in several human secretions, e.g., throat, saliva, pulmonary alveolar washings, and feces [6]. So far, the literature has just one publication showing the isolation of SARS-CoV-2 in urine; on it, the authors injected the isolated virus on culture of human cells and were able to find viral replication. The authors speculate that urine can be one alternative pathway for the virus spreading [7] and there is only one publication, as far as we know, that has tested vaginal secretions for the presence of SARS-CoV-2 with negative results [8].

The scientists believe that the SARS-CoV-2 has animal etiology, most likely from bats or pangolins. The genetic code from the human virus subtype matches in 96% with the bat virus. Another supporting correlation of zoonotic origin is the outbreak epicenter being the Wet Animal Market in Wuhan [1].

As a new zoonotic disease, several questions still need to be answered. The previous zoonosis shows that the disease, after jumping from one species to another, does not behave as it used to do in the primary species [9]. The Zika virus [ZIKV] is one of the most recent examples. Described for the first time in 1947, ZIKV had several local outbreaks, but in 2015 made the world's headlines after infected travelers went to see the 2014 world cup matches in Brazil, spreading the disease all over the world [10,11].

The ZIKV belongs to a different family from the SARS-CoV-2, Flavivirus vs. Coronaviridae family, respectively, but as SARS-CoV-2, the ZIKV is a positive-sense single-strained RNA virus [1,12,13]. Originally transmitted through mosquito bites, the ZIKV presents unusual behavior for an arbovirus, being also transmitted by intercourse. The virus can be isolated from the male sperm up to 3 months after the first symptoms appear, and the transmission by vaginal and anal intercourse [male/female and male/male] is well documented [10].

Since it was found that the coronavirus uses the ACE-2 receptor as a pathway to enter into the cells, researchers tried isolate the virus in different tissues with these receptors. Although in previous SARS outbrake complications, such as orchitis, low sperm count was found, no previous strains from SARS-CoV were isolated from the human semen [11].

The testis has high levels of ACE-2 receptors [ACE-2r], which has led researchers to speculate about the presence of SARS-CoV-2 in semen [14,15]. Currently, there are six articles published researching the presence of SARS-CoV-2 in semen [16–21]. However, only Li D, et al. [19] have been able to isolate the virus from semen. We hope that our study will add evidence that SARS-CoV-2 is found in semen.

## 2. Methods

This is a descriptive, cross-sectional study. From May to September 2020, 15 males [>18 y.o.] with confirmed COVID-19 diagnosis through nasal swab and with mild to moderate disease, were located through our institution's database. The subjects were contacted by phone for recruitment. After agreeing to participate in the study, the subjects were instructed to masturbate after taking a full shower with soap [preventing site cross contamination], dry themselves, and ejaculate in a numbered cup provided by us. We did not require any standardized sexual abstinence period for the participating subjects that also answered a survey (Table 1). The same researcher [Author 4] contacted all subjects, and a $50.00 gift-card compensation was given to the men participating in the study.

After collected, the specimens were refrigerated under $-10\,^\circ$C until all the samples could be processed at the same time. The samples were centrifuged in our institution for 10 min under 3000 rotations per minute with the supernatant harvested and placed in a Cobas® PCR Media Tube [Roche Molecular Systems, Branchburg, NY, USA]. The tubes were shipped, at room temperature, through overnight courier to the laboratory where RT-PCR was performed. The laboratory team were experienced in SARS-CoV-2 detection in clinical samples and had been testing for the virus since the beginning of the pandemic [22].

**Table 1.** Sample Demographics and Summary.

| Subject | Symptoms | Length of Symptoms (Days) | Age [Years] | Did You Travel Outside the State in the Last Month? | Education Level | Vasectomy | ΔT Symptoms Onset to Sample Recovery (Days) |
|---|---|---|---|---|---|---|---|
| 1 | Fever, Body Aches, Chills, Vomiting, Loss of Taste, Loss of Smell | 7 to 14 | 23 | No | College | No | 4 |
| 2 | No Symptoms | 0 | 22 | No | College | No | 3 |
| 3 | Cough, Sore Throat, Loss of Taste, Loss of Smell, Difficulty Breathing, Fatigue | 7 to 14 | 22 | No | College | No | 2 |
| 4 | No symptoms | 0 | 23 | No | College | No | 5 |
| 5 | Sore Throat, Cough, Diarrhea, Sneezing, Loss of Taste, Loss of smell | to 14 | 43 | No | High School | No | 3 |
| 6 | Fever, Sore Throat | 14 to 21 | 23 | Yes | Professional School | No | 3 |
| 7 | Cough, Sore Throat, Loss of Tate, Loss of Smell | <7 | 24 | Yes | Professional School | No | 4 |
| 8 | Fever, Sore Throat, Fatigue | 14 to 21 | 23 | Yes | College | No | 4 |
| 9 | Loss of Taste, Loss of Smell | <7 | 23 | Yes | Professional School | No | 3 |
| 10 | Fever, Cough, Diarrhea, Sore Throat, Body Aches | <7 | 23 | No | College | No | 6 |
| 11 | Fever, Coughing, Diarrhea, Sore Throat | <7 | 21 | No | College | No | 4 |
| 12 | Fever, Sneezing, Body Aches, Loss of Taste, Loss of Smell | <7 | 21 | No | College | No | 7 |
| 13 | Fever, Coughing, Sneezing | <3 | 19 | No | College | No | 3 |
| 14 | Fever, Coughing, Sneezing | <3 | 19 | No | High School | No | 8 |
| 15 | Sore Throat | <3 | 20 | No | College | No | 4 |
| Mean | N/A | N/A | 23.27 | N/A | N/A | N/A | 4.2 |

The SARS-CoV-2 RNA was tested using Roche cobas 6800 [Roche Molecular Systems, Branchburg, NJ, USA] [21]. Briefly, the Cobas® PCR Media Tube was loaded onto the Roche cobas 6800 equipment where it automatically processed the nucleic acid extraction, RT-PCR setup, and the RT-PCR thermocycling and signal acquisition [no human intervention in the process]. Cobas 6800 SARS-CoV-2 test primers/probes sets target the ORF1, a nonstructural region that is specific for SARS-CoV-2, as well as the conserved, structural protein envelope E gene that is shared by the Sarbecovirus subgenus. Cobas 6800 SARS-CoV-2 tests were performed twice with 24 h of difference among the tests to confirm the diagnosis. The laboratory team mailed the results to the principal investigator team.

This study was approved by ours Institutional Review Board under the number: IRB#260951. The participating institutions provided the financial grant for the study.

## 3. Results

The mean age of our sample was 23.26 y.o. [range from 19 to 43 y.o.]. From the 15 subjects studied, two were asymptomatic [13.66%]. The duration of symptoms was less than 3 days in three patients [20%], about 1 week in five patients [30%], about 1–2 weeks in three patients [20%], and about 2–3 weeks in one patient [6.66%]. The most common symptom in our sample was fever, present in eight subjects [53.33%]. All samples were collected within a period of 2 weeks from the beginning of the symptoms, but the individual data of the samples was not collected. None of the studied subjects had a vasectomy. Four patients had traveled outside the state, but not outside the USA. The SARS-CoV-2 RNA was present in one subject [6.66%] of the studied semen (Table 1).

## 4. Discussion

Since WHO declared the COVID-19 pandemic, the knowledge about the disease's clinical, molecular, and spreading mechanisms has been growing. How the initial animal–human contamination happened is still subject to debate. Human–human spreading happens mostly from contaminated droplets' inhalation of close contacts or hand-nose/mouth seeding after touching contaminated surfaces [23,24]. The presence of the virus' RNA in sewer systems close to the hospitals and patient feces also worried the health authorities about a possible fecal–oral transmission that is not currently confirmed [25].

The ACE-2r is the SARS-CoV-2 entrance gate into the human cells. The isolation of ACE-2r in several human cells, e.g., lung, heart, esophagus, kidney, bladder, intestinal endothelium, and testicular cells [13,24], theoretically, making any cell with these receptors a target to the viral replication.

Although there is elevated ACE-2r expression in human testicular cells [15], only a few publications have tried to isolate the virus from human semen [16–21]. Aside from our study, only one other study could isolate the SARS-CoV-2 RNA in semen [17]. All studies have in common a low number of samples [from 1 to 38], and one of them focused their search exclusively on recovering patients after the acute phase of the disease [20].

In our study, we prevented some of the criticisms performed by other authors [18,26,27] on the Li, D et al. publication, i.e., 1—no available significant description of the cases [on Li D, et al. the authors cited 12 comatose or dying subjects]. It is known that higher blood viral loads in SARS-CoV-2 disease are related to disease severity [18], which makes the samples of this study presumably be composed of patients with severe cases, and may have influenced the results, increasing the chance of the virus reaching other organs and body fluids, including semen or cross-contamination. In our study, none of the patients were in the hospital environment. 2—To prevent cross-contamination of the semen sample from the patients' hands to the containers. The subjects in our series were instructed to masturbate and collect the semen samples after taking a complete body shower and drying themselves, closing the specimen cup after the ejaculation, keeping it under room temperature to be collected on the next day by our team. 3—A commercial fully automated in vitro diagnostic test approved by the FDA was used, decreasing the risk of sample contamination during the assay process and benefiting from the high sensitivity of COBAS 6800 SARS-CoV-2 test [23].

Our results show that even in a small sample size it is possible to find SARS-CoV-2 RNA in the semen, the use of two different genetic probes, and a second confirmatory test performed 24 h after the first results, which decreases the chance of false-positive by sample manipulation contamination at the laboratory.

The sexual transmission alone of SARS-CoV-2 is unlikely but possible. Human sexual interactions involve the exchange of several body fluids, e.g., saliva, respiratory droplets, or other body fluids, depending on sexual preferences, that already have their role established in SARS-CoV-2 spreading [7,8,14,25]. Since the number of cases is still growing, the health authorities should be alert to possible new forms of transmission [2].

One weakness of our study is that with the small number of positive samples, the follow up of the patients to determine the time of the virus sheading would not be significant. However, the literature does not support the presence of SARS-CoV-2 in semen after the acute phase of the disease [17,20], and the alterations on semen quality, fertility, and sexual transmission are at the moment only speculative [17,20,27]. Further research is still necessary to support our findings.

### 5. Conclusions

We demonstrated that SARS-CoV-2 can be found in semen; however, the present data suggest that it is not common. So far, only our study and Li D et al. isolated the SARS-COV-2 RNA in the semen. Both studies researched patients in the disease's active phase, finding a prevalence between 6.66% to 15.8% of SARS-CoV-2 in semen, respectively. Even with a low number of samples, the studies were able to find the virus RNA in the human semen, which warrants future research to determine the clinical significance of these findings.

**Author Contributions:** B.M.—Concept, IRB submission, Data analisys, Article Writing and Revision. G.B.B.—PCR execution, Data analisys, Article Writing. N.S.—IRB Submission, Sample Recovery, Article Revision. J.M.—Patient Contact, Sample recovery, Article Writing. H.d.S.L.—Concept, Data analysis. T.H.S.R.—PCR execution, Data Analysis. R.D., R.H.J.—Funding Support. All authors have read and agreed to the published version of the manuscript.

**Funding:** No external funding.

**Institutional Review Board Statement:** IRB#260951.

**Informed Consent Statement:** Waived, since the that the fact that the patients gave us the sample it was considered sufficient to the IRB committee as the patient consent.

**Data Availability Statement:** Not applicable.

**Acknowledgments:** The Authors would like to express their appreciation to Jon Blevins and team for their technical support in the execution of this research.

**Conflicts of Interest:** The authors declare no conflict of interest.

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
