# Peer review of "Presence of SARS-CoV-2 RNA in Semen—Cohort Study in the United States COVID-19 Positive Patients"

_2036-7449, doi:10.3390/idr13010012_

Round 1

Reviewer 1 Report

The paper reads well, even though it is quite repetitive and needs quite a bit English revision. 

I would recommend the authors add a couple of figures describing their data - simple piecharts for the proportion of individuals with/ without SARS-CoV2 in the semen, age and symptoms.

I would also like to see a comparison of other coronaviruses found in semen, and not only Zika virus. 

It is interesting that other in silico screening predicted the presence of SARS-CoV2 in testis[1], and some other papers actually show that SARS-CoV2 could be found in testis [2]. 

It lacks a review of the literature regarding other studies that found or not SARS-CoV2 in semen[3,4] and many others. 

  1. Gysi, D. M., Do Valle, Í., Zitnik, M., Ameli, A., Gan, X., Varol, O., … Barabási, A.-L. (2020). Network Medicine Framework for Identifying Drug Repurposing Opportunities for COVID-19. ArXiv.
  2. Fan, C., Lu, W., Li, K., Ding, Y., & Wang, J. (2021). ACE2 Expression in Kidney and Testis May Cause Kidney and Testis Infection in COVID-19 Patients. Frontiers in Medicine, 7, 563893. https://doi.org/10.3389/fmed.2020.563893

  3. Massarotti, C., Garolla, A., Maccarini, E., Scaruffi, P., Stigliani, S., Anserini, P., & Foresta, C. (2021). SARS‐CoV‐2 in the semen: Where does it come from? Andrology, 9(1), 39–41. https://doi.org/10.1111/andr.12839

  4. Paoli, D., Pallotti, F., Turriziani, O., Mazzuti, L., Antonelli, G., Lenzi, A., & Lombardo, F. (2021). SARS‐CoV‐2 presence in seminal fluid: Myth or reality. Andrology, 9(1), 23–26. https://doi.org/10.1111/andr.12825

Author Response

Dear Revisor,

We appreciate the time you have spent in reviewing our article,

One of the authors performed a style revision, a native speaker, and we expect that the article is more comprehensive and easy to read.

We redrew the chart to make it more appealing.

When performing the literature review, we came across with one of your suggested articles:

Massaroti Cet al -  SARS‐CoV‐2 in the semen: Where does it come from?

However interesting, it is an opinion article.

Paoli D et al. - SARS‐CoV‐2 presence in seminal fluid: Myth or reality. is reference 27 in or article

We did not cite the coronavirus's isolation in the testis because the articles that detected it cited the findings are made post-Morten necropsies. Even being interesting information, the examinations post-Morten were out of the scope from our article. 

We add a paragraph comparing the presence of other coronaviruses in semen and SARS-CoV-2 as suggested by you.

We hope the explanations above and the version of our article are more suitable for your view.

Best Regards,

The Authors.

Reviewer 2 Report

It is not clear the mean time between the swab positivity for SARS-CoV-2 and the sample recovery of the semen. This must be explicited to understand the intervall from the theoretically acute fase (even if asymptomatic). This is even more important for asymptomatic cases (n.2,4).

Some authors report that viral spread in the stool is late. So I advice to add a sentence underlining the limitations of this paper, saying that we don't know so far the timing of the viral shedding in the semen (if confirmed from other studies).

The final sentence in the discussion (" Perhaps we will find the virus... quality") is completely speculative without supporting data. I suggest to delete it.

Author Response

Dear Revisor, 

We appreciate the time you invested in the review of our article.

Unfortunately, we don't have the individual data from each sample collection and the begging test. According to the hospital HIPPA regulations the results of the patients eligible for the study were sent to us every Tuesday and Friday by the gatekeepers, which made us be able to contact the patient in a maximal interval of 2 weeks since the nasopharyngeal test was performed.

We added a sentence about the limitations as you have suggested

We deleted the phase as you have suggested

Once again we appreciate our time

Regards,

The Authors

Reviewer 3 Report

It remains uncertain if this is the first study on the detection of SARS-CoV-2 in urine because the authors cannot rule out the possibility of the presence of the virus in urine. Therefore, I suggest the following revisions to be made.

Line 48: While a number of studies do not show the presence of SARS-CoV-2 in urine (e.g. doi: 10.1001/jama.2020.3786, 10.1002/jmv.25825, 10.1016/S1473-3099(20)30196-1, and 10.1001/jama.2020.3204), one has shown possible presence doi: 10.1080/22221751.2020.1760144 and the authors should discuss this work. It is possible that the detection of SARS-CoV-2 in semen is related to the urine.

Line 158: “We prove” sounds too strong and should be rephrased.

Certain information in square brackets should be stated clearly, line 102 for instance. Is “19 to 43” the range or the IQR of age?

Author Response

Dear revisor, 

We appreciate the time you invested in reviewing our article. 

As you suggested, we changed the wording of our conclusion and added the article about the SARS-CoV-2  being isolated on the urine. Our original article was written before the publication of the cited article. 

We also made clear the age information as you have requested.

The English revisions were made for one of the Authors who is a native speaker, and we believe now that the ideas are exposed more comprehensively.

We hope the modifications meet your expectations.

Sincerely,

The Authors

Round 2

Reviewer 1 Report

Thank you. All my suggestions in the main text were addressed. 

Reviewer 3 Report

I have no further comments.